# Prevalence and Risk Factors for Superinfection with a Difficult-to-Treat Pathogen in Periprosthetic Joint Infections

**DOI:** 10.3390/antibiotics14080752

**Published:** 2025-07-25

**Authors:** Ali Darwich, Tobias Baumgärtner, Svetlana Hetjens, Sascha Gravius, Mohamad Bdeir

**Affiliations:** 1Department of Orthopedic and Trauma Surgery, University Medical Center Mannheim, Medical Faculty Mannheim, University of Heidelberg, Theodor-Kutzer-Ufer 1-3, 68167 Mannheim, Germany; ali.darwich@umm.de (A.D.); tobias.baumgaertner@umm.de (T.B.); 2Institute of Medical Statistics and Biomathematics, University Medical Center Mannheim, Medical Faculty Mannheim, University of Heidelberg, Theodor-Kutzer-Ufer 1-3, 68167 Mannheim, Germany; svetlana.hetjens@medma.uni-heidelberg.de

**Keywords:** risk factors, prevalence, difficult to treat, pathogen, microorganism, superinfection, periprosthetic joint infection

## Abstract

**Background:** Periprosthetic joint infections (PJIs) are considered as one of the most serious complications after total joint arthroplasty. Aim of this study was to evaluate the prevalence of PJI caused by difficult-to-treat (DTT) pathogens as well as PJIs with a superinfection with a DTT pathogen in the course of the infection and assess the risk factors leading to this emergence. **Methods:** Data of 169 consecutive patients with a PJI was analyzed in this retrospective observational single-center study, and cases were categorized into PJIs with initial DTT pathogens, PJIs with DTT pathogen superinfection, non-DTT PJIs, and PJIs with superinfection. Recorded parameters comprised age, gender, side, body mass index (BMI), preoperative anticoagulation, and serum level of C-reactive protein (CRP) at admission, as well as preoperative patient status using the ASA (American Society of Anesthesiologists) score and the age-adjusted form of the Charlson comorbidity index (CCI). Furthermore, the infecting microorganism and the type of infection as well as the chosen operative treatment regime, duration of the antibiotics interval, and the outcome were recorded. **Results:** In total, 46.2% of cases were DTT PJIs, and 30.8% of them were superinfections. Elevated serum CRP levels at admission (≥92.1 mg/L) were linked to a nearly 7-fold increased likelihood of a DTT PJI (OR 6.981, CI [1.367–35.63], *p* = 0.001), compared to patients with a non-DTT PJI. Hip joint involvement was also associated with a 3.5-fold higher risk compared to knee joints (OR 3.478, CI [0.361–33.538], *p* = 0.0225). Furthermore, patients undergoing ≥3 revision surgeries demonstrated a significantly 1.3-fold increased risk of developing a DTT superinfection (OR 1.288, CI [1.100–1.508], *p* < 0.0001). Chronic PJIs were similarly associated with a markedly 3.5-fold higher likelihood of superinfection by DTT pathogens (OR 3.449, CI [1.159–10.262], *p* = 0.0387). Remaining parameters did not significantly affect the rate of a DTT PJI or a PJI with DTT superinfection. **Conclusions:** These findings underscore the importance of early identification of high-risk patients and highlight the need for tailored preventive and therapeutic strategies in managing DTT PJIs.

## 1. Introduction

Total joint arthroplasty (TJA) is one of the most successful and most commonly performed surgical interventions in orthopedics due to its improvement of quality of life and pain relief in patients with joint disease [1]. Despite this success, periprosthetic joint infection (PJI) remains a serious complication, involving 1% to 2% of primary total hip and knee joint replacement cases [2] and up to 15% of revision cases [3]. In the literature, surgical and microbiological failure rates of up to 40% after one- or two-stage revisions are reported [4,5].

Due to its substantial impact on the clinical end outcome and on the healthcare system, a PJI and its management have become a great challenge to both infectiologists and orthopedic surgeons [6].

One of the aspects determining the outcome in the context of a PJI is the infecting pathogen and its resistance profile [7,8]. A PJI caused by resistant pathogens such as methicillin-resistant *Staphylococcus aureus* [9], *Enterococci*, and extended-spectrum beta-lactamase-producing *Enterobacterales* [10], Gram-negative microorganisms [11], and yeasts [12] are associated with poor outcomes. An additional important factor on the course of the PJI is the biofilm activity of the infecting pathogen [13,14].

Wimmer et al. [15] defined difficult-to-treat (DTT) microorganisms as those for which no effective, biofilm-active, and highly bioavailable bactericidal antibiotics are available. This includes pathogens that are only susceptible to intravenous antibiotics, such as fluoroquinolone-resistant *Pseudomonas* spp. or azole-resistant yeasts, due to the absence of suitable oral alternatives [15]. Similarly, organisms sensitive only to antibiotics with limited oral bioavailability, such as *Staphylococcus* spp. resistant to trimethoprim/sulfamethoxazole, doxycycline, or linezolid, and *Enterococcus* spp. resistant to aminopenicillins, were also classified as DTT [15]. Rifampin and fluoroquinolones are known for their potent biofilm activity and high oral availability. *Staphylococcus* species resistant to these antibiotics were specifically considered DTT as well. Moreover, slow-growing bacteria such as small colony variants were included in the DTT category due to their enhanced ability to persist intracellularly [16] and form biofilms [17], traits that enable them to evade antimicrobial therapies.

High rates of treatment failure were associated with PJIs caused by DTT pathogens (DTT PJIs) [18,19,20].

The previous study by Darwich et al. showed the poorer outcome of a PJI with DTT superinfection compared to a PJI with non-DTT pathogen or a PJI with initial DTT pathogen without superinfection [21]. Therefore, the aim of this study was to define the risk factors that contribute to the development of DTT superinfection in the course of infection and to evaluate the prevalence of the PJI caused by DTT pathogens as well as the PJI with a superinfection with a DTT pathogen.

## 2. Results

In this study 169 patients with 169 infected prostheses were included. The group of patients with a PJI caused by a non-DTT pathogen (PJI non-DTT) consisted of 91 patients (53.8%). A total of 78 patients presented with a PJI involving a DTT pathogen (PJI DTT total) (46.2%); in 54 of them (69.2%), the DTT pathogen was detected in the first operation (PJI DTT initial), and in 24 patients (30.8%) the DTT pathogen was considered as a superinfection (PJI DTT superinfection) and was detected at a later stage in the course of the PJI. The total number of superinfections involving either a DTT or a non-DTT pathogen (PJI superinfection total) was 40 patients (23.7%).

There were no significant differences in the patient parameters recorded between groups. Table 1 shows an overview of the demographic data of all patients as well as of the formed groups.

The mean number of surgical revisions in the cohort as a whole was 3.3 ± 3 revisions. In comparison to a PJI involving DTT pathogens (PJI DTT total), a non-DTT PJI showed significantly lower number of surgical revisions with 2.3 ± 1.9 versus 4.4 ± 3.5 revisions (*p* < 0.0001). Among the PJI involving DTT pathogens (PJI DTT total), the PJI with DTT superinfections (PJI DTT superinfection) showed the highest number of surgical revisions with a mean revision rate of 6 ± 3.6 revisions (*p* < 0.0001). This revision rate was even higher than that of the group of patients with the PJI involving a DTT pathogen from the start of the infection without a superinfection (PJI DTT initial), where a mean revision rate of 3.7 ± 3.3 revisions (*p* < 0.0001) was observed. In the group with the PJI with DTT superinfection (PJI DTT superinfection), 2.4 ± 1.3 revisions were made before the DTT pathogen was detected.

Regarding antibiotic treatment, a mean duration of 54.8 ± 36.7 days was observed in the cohort as a whole. In comparison to the PJI DTT total, the non-DTT PJI showed significantly lower durations of antibiotic treatment with 46.3 ± 29.5 days versus 64.2 ± 41.4 days (*p* = 0.0495). In the total cohort, the group of the PJI DTT superinfection showed the longest total duration of antibiotic treatment, with a mean of 71.2 ± 45.2 days (*p* = 0.0023). Concerning therapeutic concepts, similar results were observed: The highest rate of prosthesis exchange regime without cement spacer implantation was documented in the group with PJI DTT superinfection, with 24% of cases (*p* = 0.0080). In 10 of the 24 patients with a DTT superinfection (41.7%), the surgical treatment concept was changed after detection of the DTT pathogen. All perioperative data are listed in Table 2.

The most detected DTT pathogens in both groups with PJI DTT initial and PJI DTT superinfection were coagulase-negative *Staphylococci*, followed by *Enterococci*. Three cases of polymicrobial infection each with a combination of a coagulase-negative *Staphylococcus* and an *Enterococcus* were detected. The differences observed in the distribution of microorganisms between the PJI groups were not statistically significant. The DTT pathogen distribution is shown in Table 3.

Regarding risk factors associated with PJI involving DTT pathogens (PJI DTT total), a C-reactive protein (CRP) value at admission of ≥92.1 mg/L was associated with approximately a 7-fold increase in the likelihood of detecting a DTT PJI (OR 6.981, CI [1.367–35.63], *p* = 0.001). The hip joint was a further risk factor with approximately a 3.5-fold increase in the likelihood of detecting a DTT PJI (OR 3.478, CI [0.361–33.538], *p* = 0.0225). Concerning risk factors associated with the development of a superinfection with a DTT pathogen in the course of the PJI (PJI DTT superinfection), the type of infection and the number of revisions were recognized as risk factors. Chronic infections were associated with approximately a 3.5-fold increase in the likelihood of developing a superinfection with a DTT pathogen during the PJI (OR 3.449, CI [1.159–10.262], *p* = 0.0387). Similarly, patients with ≥3 revisions were associated with approximately a 1.3-fold increase in the likelihood of developing a superinfection with a DTT pathogen (OR 1.288, CI [1.100–1.508], *p* < 0.0001).

With respect to sociodemographic characteristics including age, sex, ASA (American Society of Anesthesiologists) score, surgical and antibiotic treatment, body mass index (BMI), and anticoagulation status, the study groups were comparable, and no statistically significant differences were observed. A summary of the assessed risk factors is presented in Table 4.

In this study, the follow-up period was 13.5 ± 10.8 months (1–112). The median infection-free survival for the PJI non-DTT was 450 days (95% CI: 390–540), for the PJI superinfection non-DTT 450 days (95% CI: 360–540), for the PJI DTT initial 520 days (95% CI: 360–930), and for the PJI DTT superinfection 540 days (95% CI: 360–630). Seven patients could no longer be followed as they live abroad. Twelve patients were dead due to a non-PJI-related cause. The Kaplan–Meier curves for infection-free survival of the studied groups are presented in Figure 1.

## 3. Discussion

In a continuously aging society, the number of joint replacements is steadily increasing. At the same time, the average age of patients receiving arthroplasties is decreasing, leading to a continuous rise in the number of periprosthetic joint infections (PJIs) worldwide. PJIs remain a major challenge for medical personnel and place a significant burden on patients. Difficult-to-treat (DTT) PJIs represent a particularly challenging subgroup, where knowledge and the available literature are especially limited [22,23].

Several studies have investigated the various risk factors associated with an increased risk of a PJI. Bozic et al. identified preexisting rheumatologic disease, obesity, coagulopathy, and preoperative anemia as risk factors [24]. Mangram et al. and Poultsides et al. reported that immune disorders, diabetes, poor nutritional status, and smoking contribute to an elevated risk of a PJI in hip and knee arthroplasty. Advanced age and male gender have also been associated with a higher risk [25]. In contrast, the literature on risk factors specifically associated with a higher risk of a DTT PJI is scarce or even nonexistent.

Serum CRP is an established and valuable marker in the diagnosis of a PJI. The MSIS guidelines recognize high serum CRP as one of the indicators for diagnosing a periprosthetic joint infection [26]. According to several studies, serum CRP is not inferior to synovial CRP in the diagnosis process of a PJI [27,28]. A recent study by Grzelecki et al. demonstrated high sensitivity and specificity of serum CRP as a diagnostic marker for a PJI, with values of 84.9% and 90.5%, respectively. In the subgroup of patients with a high-virulence PJI, the sensitivity of CRP was even higher, reaching 94.8%. [29].

In the present study, a high serum CRP level at admission was identified as a risk factor for a DTT PJI. CRP values above 92.1 mg/L were associated with approximately a 7-fold increase in the likelihood of detecting the DTT PJI (OR 6.981, CI [1.367–35.63], *p* = 0.001), compared to patients with the non-DTT PJI. This finding aligns with the observations of Unter Ecker et al. [30], who reported significantly higher mean serum CRP levels in the PJI group compared to patients undergoing aseptic revision (50.2 mg/L vs. 11.6 mg/L). Additionally, Unter Ecker et al. found that CRP levels were significantly higher in PJIs caused by high-virulence pathogens than in those caused by low-virulence microorganisms in both hip and knee joints (*p* < 0.0001). Similarly, Ghanem et al. found significantly higher mean CRP values in PJI patients compared to those undergoing aseptic revision, with values of 14.9 mg/L versus 9.48 mg/L (*p* < 0.0001) [31]. Grossi et al. investigated patients with a DTT PJI involving Gram-negative bacilli and observed that serum CRP levels of ≥175 mg/L were a risk factor associated with treatment failure [32]. Baertl et al. conducted a study comparing patients with a PJI who developed PJI-related sepsis during the infection and found significantly higher CRP values in the group of patients with a hip PJI who developed sepsis, with 130.0 mg/L versus 77.0 mg/L (*p* = 0.017). The authors concluded that elevated serum CRP levels were associated with an increased risk of developing sepsis as well as with higher rates of persistence and recurrence of the PJI [33].

In the present study, the affected joint was also identified as a risk factor for the DTT PJI. Hip joints were associated with approximately a 3.5-fold increase in the likelihood of detecting a DTT PJI (OR 3.478, CI [0.361–33.538], *p* = 0.0225) compared to knee joints. These results are consistent with findings by Bae et al. [34], who investigated PJI rates in 12,320 primary total joint arthroplasties and reported PJI rates of 0.33% in hip joints versus 0.22% in knee joints. Tsai et al. [35] investigated 294 cases of PJIs and observed that polymicrobial PJIs, anaerobic PJIs, and PJIs caused by enteric Gram-negative bacilli were significantly more common in hip joints than in knee joints, with 22 vs. 6 cases (*p* = 0.006), 11 vs. 0 cases (*p* = 0.002), and 20 vs. 6 cases (*p* = 0.014), respectively. Similarly, Enz et al. [36] examined a DTT PJI caused by fungi and found that hip joints were significantly more frequently affected than knee joints (77.8% vs. 22.2%, *p* = 0.045). Tornero et al. [37] examined 203 cases of an enterococcal PJI across 18 European centers between 1999 and 2012, with 63% of infections affecting the hip joint. *Enterococcus faecalis* was the most isolated species, and up to 33% of isolates were classified as a DTT vancomycin-resistant *Enterococci* (VRE) PJI.

Darwich et al. [21] reported in a previous study that PJIs initially caused by non-DTT pathogens that subsequently developed a superinfection or shifted to a DTT pathogen during the course of infection were associated with significantly worse clinical outcomes. These outcomes were poorer not only in comparison to PJIs without any DTT involvement but also compared to infections initially caused by DTT pathogens without superinfection. This highlights the critical impact of secondary DTT involvement on prognosis.

In the current study, the number of revision surgeries was identified as a significant risk factor for developing a superinfection with a DTT pathogen during the course of a PJI. Patients who underwent ≥ 3 revisions showed an approximately 1.3-fold increased likelihood of acquiring such a superinfection (OR 1.288, CI [1.100–1.508], *p* < 0.0001). These findings are consistent with those of Baumgärtner et al. [38], who investigated DTT PJIs caused by rifampin-resistant organisms and observed a significantly higher number of revision surgeries in rifampin-resistant cases compared to rifampin-sensitive ones (6.9 ± 5.1 vs. 3.59 ± 3.39 revisions, *p* = 0.0011). Achermann et al. [39] investigated rifampin resistance in a staphylococcal PJI and identified a history of ≥3 surgical revisions as a risk factor for developing rifampin-resistant DTT PJIs. Similarly, Wimmer et al. [15] found that patients with DTT PJIs required significantly more revision surgeries than those with non-DTT PJIs (4.72 ± 3.03 vs. 2.41 ± 3.02 operations, *p* < 0.05), further underscoring the association between surgical burden and DTT infections.

Chronic type of PJI was also identified as a risk factor for developing a superinfection with a DTT pathogen. Chronic infections were associated with approximately a 3.5-fold increase in the likelihood of developing such a superinfection during the PJI (OR 3.449, CI [1.159–10.262], *p* = 0.0387). This aligns with the findings of Baumgärtner et al. [38], who investigated rifampin resistance in PJIs and observed that 75% of rifampin-resistant DTT PJIs were chronic cases, compared to 34.3% in the group with rifampin-sensitive PJIs (*p* = 0.0013).

The two identified risk factors for the development of a DTT superinfection, namely, a higher number of revision surgeries and chronic infection, should not be viewed in isolation. As described under the subsection “Section 4.4” in the Section 4, a two- or multi-stage exchange protocol was employed for managing chronic PJIs. This may help explain the observed association, as chronic infections inherently involve more complex and prolonged treatment courses, often requiring multiple revisions, which in turn may increase the risk for DTT superinfections.

DTT PJIs often lead to prolonged hospitalizations, multiple surgeries, and compromised patient outcomes. Managing such devastating infections requires a multidisciplinary approach, with close collaboration between orthopedic surgeons and infectiologists.

Preventive strategies begin with careful preoperative planning. This includes managing obesity [40], encouraging cessation of tobacco use [41], and optimizing chronic comorbid conditions such as diabetes mellitus (with strict glycemic control) [42], all of which have been associated with an increased risk of a PJI [2].

Intraoperatively, multiple strategies should be considered. These include general measures, such as optimizing surgical site skin preparation protocols [43] and ensuring strict adherence to operating room environmental controls, including sterilization practices and appropriate airflow systems [44], as well as more specific interventions such as antibiotic prophylaxis, with particular attention to the early initiation of broad-spectrum antibiotics in high-risk patients [45].

Postoperative care is equally important. Closer monitoring of high-risk patients, as identified in this study (e.g., those with chronic infections, undergoing multiple revisions, or presenting with elevated CRP levels), may help facilitate early detection and intervention in cases of emerging superinfections.

Moreover, some authors advocate for the direct use of a two-stage exchange procedure in cases of a confirmed or suspected DTT PJI, given the high failure rates associated with single-stage procedures in this context [15].

Based on the findings of the current study, clinical practice may benefit from risk-adapted treatment strategies. Specifically, patients presenting with high CRP levels and hip PJIs, both of which were identified as significant risk factors for the presence or development of a DTT PJI and DTT superinfection and should be considered for broader empirical antibiotic coverage. In addition, these patients may warrant earlier consideration of a two-stage exchange procedure rather than attempting debridement and implant retention, particularly in settings where superinfection or DTT pathogens are suspected. Such an approach may improve outcomes by addressing the infection more aggressively and tailoring management to the elevated risk profile, thereby enhancing the clinical applicability of the study’s results. This is consistent with the recommendations of Osmon et al., who, as part of a working group of the Infectious Diseases Society of America (IDSA), published clinical practice guidelines for the diagnosis and management of PJIs. In cases of PJIs caused by DTT organisms, the panel specifically advised a two-stage exchange procedure [46].

One of the limitations of this study is the retrospective and descriptive design, resulting in a low level of evidence. Although the study is a single-center study with a prolonged follow-up that reports on a cohort larger than similar prior studies, the relatively small overall sample size constitutes a limitation. The choice of the treatment regimen in the analyzed cases represents another limitation. These cases were included over a period of 5 years. Due to this long duration of inclusion, the therapy regimes were obviously subject to further development and improvement based on the evolving and available literature, which may have had a minor effect on the course of the infection and created heterogenicity in the applied therapy protocols.

The surgical treatment of the included cases was performed by different surgeons who may have used different surgical techniques. Despite the application of the same well-defined internal algorithm in the treatment, a small effect on the course of the infection cannot be excluded, which could be seen as another limitation of this study.

## 4. Materials and Methods

### 4.1. Study Population

The routine clinical data of all patients presenting to our university hospital over a period of five years with an acute or chronic PJI were collected and in the context of an observational single-center study retrospectively analyzed. Patients undergoing follow-up for a PJI who showed no microbial growth in cultures were however excluded from the study.

### 4.2. Patient Parameters

The following demographic parameters have been recorded: age, gender, side, body-mass index (BMI), preoperative anticoagulation, and the serum level of C-reactive protein (CRP) at admission. The ASA (American Society of Anesthesiologists) score [47] and the age-adjusted form of the Charlson comorbidity index (CCI) were used for the evaluation of the preoperative patient status. Based on the classification suggested by Lunz et al., patients were categorized according to their age-adjusted CCI in several groups: CCI 0 to 1, CCI 2 to 3, CCI 4 to 5, and CCI above 5 [48,49,50]. It was also recorded if the included patients received any antimicrobial or surgical treatment before admission.

Furthermore, the infecting microorganism and the chosen operative treatment regime, duration of the antibiotics interval, and the outcome were recorded. According to the guidelines of the Infectious Diseases Society of America, acute infections were defined as those that occur within 4 weeks of the initial implantation or present with symptoms lasting less than 3 weeks. Infections that fall outside of these timeframes were classified as chronic [46].

### 4.3. Definition of Periprosthetic Joint Infection and DTT Pathogens

A PJI was defined and diagnosed according to the criteria of the Musculoskeletal Infection Society (MSIS) [51] and the modifications proposed by Parvizi et al. [26]. As mentioned before, pathogens for which no highly bioavailable biofilm-active antibiotic is available were categorized as DTT. As defined by Wimmer et al. [15], this category included resistant pathogens sensible only to agents with low oral bioavailability or to agents only available for intravenous use as well as pathogens with resistance to highly biofilm-active medication such as rifampin and fluoroquinolones. “Superinfection” was defined as either the emergence of a completely new pathogen during the course of infection that meets the DTT criteria described above or the development of resistance in an existing pathogen, thereby resulting in its classification as a DTT pathogen.

The following PJI groups were defined:-PJI caused by a non-DTT pathogen without superinfection.-PJI caused by a DTT pathogen without superinfection.-PJI initially caused by a non-DTT pathogen but with subsequent superinfection or pathogen switch to a DTT pathogen later in the course of the infection.-PJI with superinfection of any kind (DTT and non-DTT pathogens).

### 4.4. Treatment Regimen

The most appropriate surgical treatment was individually chosen according to a standardized algorithm based on earlier publications of Zimmerli et al. [52,53]. Surgical treatment options included debridement and exchange of the mobile parts with prosthesis retention or two-stage prosthesis exchange with or without dual antibiotic-loaded bone-cement spacer implantation. Available combination for dual loading were 0.5 g gentamicin and 2 g vancomycin per 40 g PMMA (Polymethylmethacrylate) or 1 g gentamicin and 1 g clindamycin per 40 g PMMA. The suitable bone cement was chosen based on the identified microorganisms in previous surgeries [54].

Generally, antibiotics were administered intravenously for the first 2 weeks postoperatively, then orally for 4 weeks under close clinical and laboratory monitoring.

Satisfactory local wound situation without swelling, erythema, tenderness, or discharge as well as laboratory results without any sign of infection (CRP < 10 mg/L) following a 2-week antibiotic-free interval were a prerequisite for the reimplantation [55]. A preoperative joint aspiration was not routinely performed. In case clinical and laboratory signs of infection were present, a further surgical revision with additional debridement and new intraoperative specimen collection as well as spacer exchange (if applicable) was performed.

A similar antimicrobial treatment protocol was followed after reimplantation with intravenous antibiotics for 2 weeks followed by biofilm-active antimicrobials with high oral bioavailability for 4 weeks. During this phase close clinical and laboratory monitoring was carried out [56].

### 4.5. Statistical Analysis

For all statistical analysis, SAS software, release 9.4 (SAS Institute Inc., Cary, NC, USA), was used. For patients’ parameter and perioperative data, binary variables are presented as percentages of patients per characteristic and continuous variables as mean values with standard deviations or median values. The chi-square test or Fisher’s exact test were used in order to compare groups with respect to qualitative parameters, asappropriate. Adjustment for multiple comparisons was performed using the Scheffé test. ANOVA was used to compare several groups. For normally distributed samples of continuous variables, the *t*-test was used. For non-normally distributed samples, or when the requirements for the *t*-test were not met, the Mann–Whitney U test was used. Logistic regression analysis was employed for testing the association between a binary outcome and an explanatory quantitative variable. Furthermore, the optimum cutoff point was estimated, which minimizes the Youden index (sensitivity + specificity −1). A *p*-value of less than 0.05 was considered as statistically significant.

### 4.6. Ethics Approval

This study was approved by the Ethics Committee of clinical research at our institution (Ethics Committee II, University Medical Center Mannheim, Medical Faculty Mannheim, Heidelberg University, Theodor-Kutzer-Ufer 1-3, 68167, Mannheim, Approval 2021-814) and performed in accordance with the local ethical standards and the principles of the 1964 Helsinki Declaration and its later amendments.

## 5. Conclusions

This study identified several significant risk factors associated with DTT PJIs. Elevated serum CRP levels at admission (≥92.1 mg/L) were linked to a nearly 7-fold increase in the likelihood of DTT PJIs. Hip joint involvement was also associated with a 3.5-fold higher risk compared to knee joints. Furthermore, patients undergoing ≥3 revision surgeries demonstrated a significantly 1.3-fold increased risk of developing a DTT superinfection. Chronic PJIs were similarly associated with a markedly 3.5-fold higher likelihood of superinfection by DTT pathogens. These findings underscore the importance of early identification of high-risk patients and highlight the need for tailored preventive and therapeutic strategies in managing DTT PJIs.

## Figures and Tables

**Figure 1 antibiotics-14-00752-f001:**
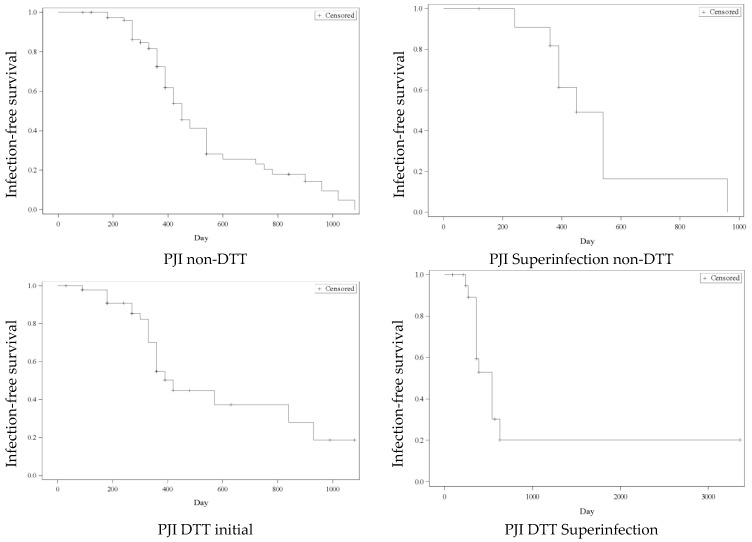
Kaplan–Meier curves for infection-free survival of the studied groups.

**Table 1 antibiotics-14-00752-t001:** Patient parameters.

		Age (Years)	Sex		Side		BMI (Kg/m^2^)	ASA Score	Anticoagulation	
		Mean ± SD (Range)	*N* (%)		*N* (%)		Mean ± SD (Range)	Mean ± SD	*N* (%)	
			Female	Male	Right	Left			Yes	No
Total	*n* = 169	71.1 ± 13.1 (20–97)	84 (49.7%)	85 (50.3%)	81 (47.9%)	88 (52.1%)	29.5 ± 6.5 (16.4–46.8)	3 ± 0.6	67 (39.5%)	102 (60.5%)
PJI non-DTT	*n* = 91	70.6 ± 13.9 (20–93)	47 (51.7%)	44 (48.3%)	38 (41.8%)	53 (58.2%)	28.8 ± 6.5 (17.6–46.8)	3 ± 0.6	31 (34.1%)	60 (65.9%)
PJI DTT total	*n* = 78	71.5 ± 12 (35–97)	37 (47.4%)	41 (52.6%)	41 (52.6%)	37 (47.4%)	30.3 ± 6.4 (16.4–45.1)	3 ± 0.6	37 (47.4%)	41 (52.6%)
PJI DTT initial	*n* = 54	71.8 ± 10.9 (48–89)	28 (51.9%)	26 (48.1%)	26 (48.1%)	28 (51.9%)	30.7 ± 6.3 (16.4–45.1)	3 ± 0.6	24 (44.4%)	30 (55.6%)
PJI DTT super-infection	*n* = 24	71.1 ± 14.6 (35–97)	9 (37.5%)	15 (62.5%)	15 (62.5%)	9 (37.5%)	29.5 ± 6.7 (18.5–42.3)	3 ± 0.8	13 (54.2%)	11 (45.8%)
PJI super-infection total	*n* = 40	69.1 ± 13.3 (35–97)	15 (37.5%)	25 (62.5%)	20 (50%)	20 (50%)	30.4 ± 6.9 (18.5–44.1)	3 ± 0.7	15 (37.5%)	25 (62.5%)
*p*-value *		0.6809	0.5898		0.2663		0.3324	0.7147	0.0743	

PJI, periprosthetic joint infection; DTT, difficult to treat; SD, standard deviation; BMI, body mass index; ASA, American Society of Anesthesiologists score; * *p* < 0.05 indicates statistical significance.

**Table 2 antibiotics-14-00752-t002:** Perioperative data.

		Affected Joint			Type of Infection		CCI Age Adjusted			CRP (mg/L) at Admission	Antibiotic Treatment	Treatment Before Admission		Surgical Treatment			Number of Revisions
		*N* (%)			*N* (%)		*N* (%)			Mean ± SD (Range)	Mean ± SD (range)	*N* (%)		*N* (%)			Mean ± SD (Range)
		Hip	Knee	Shoulder	Acute	Chronic	0–1	2–3	4–5		Total Duration in Days	Antibiotic	Surgical	Prosthesis Retention	Prosthesis Exchange with Cement Spacer	Prosthesis Exchange Without Cement Spacer	
Total	*n* = 169	83 (49.1%)	75 (44.4%)	11 (6.5%)	73 (43.2%)	96 (56.8%)	9 (5.3%)	108 (63.9%)	52 (30.8%)	77.4 ± 83 (2–386)	54.8 ± 36.7 (4–228)	24 (14.2%)	25 (14.8%)	66 (40.7%)	75 (46.3%)	21 (13%)	3.3 ± 3 (1–20)
PJI non-DTT	*n* = 91	36 (39.6%)	47 (51.6%)	8 (8.8%)	33 (36.3%)	58 (63.7%)	6 (6.6%)	60 (65.9%)	25 (27.5%)	105.6 ± 101.6 (2–390)	46.3 ± 29.5 (4–217)	13 (14.3%)	11 (12.1%)	44 (52.4%)	33 (39.3%)	7 (8.3%)	2.3 ± 1.9 (1–14)
PJI DTT total	*n* = 78	47 (60.3%)	28 (35.9%)	3 (3.8%)	37 (47.4%)	41 (52.6%)	3 (3.8%)	48 (61.5%)	27 (34.6%)	64.4 ± 81.8 (2–381)	64.2 ± 41.4 (4–228)	11 (14.1%)	14 (17.9%)	22 (28.2%)	42 (53.8%)	14 (18%)	4.4 ± 3.5 (1–20)
PJI DTT initial	*n* = 54	34 (63%)	20 (37%)	0 (0%)	22 (40.8%)	32 (59.2%)	1 (1.9%)	36 (66.7%)	17 (31.5%)	51.86 ± 63.01 (2–381)	61.1 ± 39.7 (4–181)	6 (11.1%)	6 (11.1%)	15 (28.8%)	31 (57.4%)	8 (14.8%)	3.7 ± 3.3 (1–20)
PJI DTT Superinfection	*n* = 24	13 (54.2%)	8 (33.3%)	3 (12.5%)	15 (62.5%)	9 (37.5%)	2 (8.3%)	12 (50%)	10 (41.7%)	92.7 ± 109.7 (2–370)	71.2 ± 45.2 (14–228)	5 (20.8%)	5 (20.8%)	7 (30.2%)	11 (45.8%)	6 (24%)	6 ± 3.6 (2–14)
PJI Superinfection total	*n* = 40	17 (42.5%)	17 (42.5%)	6 (15%)	19 (47.5%)	21 (52.5%)	2 (5%)	27 (67.5%)	11 (27.5%)	92.9 ± 93.8 (2–370)	68.3 ± 45.6 (14–228)	6 (15%)	6 (15%)	11 (27.5%)	23 (57.5%)	6 (15%)	5.3 ± 3.4 (2–14)
*p*-value *		0.0025			0.3155		0.9614			0.0055	0.0023	0.8534		0.0080			<0.0001

PJI, periprosthetic joint infection; DTT, difficult to treat; SD, standard deviation; CRP, C-reactive protein; CCI, Charlson comorbidity index; * *p* < 0.05 indicates statistical significance; PJI non-DTT: PJI caused by a non-DTT pathogen without superinfection; PJI DTT total: PJI caused by a DTT pathogen with or without superinfection; PJI DTT initial: PJI caused by a DTT pathogen without superinfection; PJI DTT Superinfection: PJI superinfection caused initially by a non-DTT pathogen but with pathogen switch to a DTT pathogen later in the course of the infection. PJI Superinfection total: PJI superinfection with DTT or non-DTT.

**Table 3 antibiotics-14-00752-t003:** Identified difficult-to-treat microorganisms.

		Coagulase-Negative *Staphylococci N* (%)	*Enterococci N* (%)	*Pseudomonas aeruginosa N* (%)	*Candida albicans N* (%)	Polymicrobial * *N* (%)
PJI DTT total	*n* = 78	47 (60.3%)	25 (32.1%)	5 (6.4%)	4 (5.1%)	3 (3.9%)
PJI DTT initial	*n* = 54	34 (63%)	12 (22.2%)	3 (5.6%)	3 (5.6%)	2 (3.7%)
PJI DTT superinfection	*n* = 24	11 (45.8%)	9 (37.5%)	2 (8.3%)	1 (4.2%)	1 (4.2%)

PJI periprosthetic joint infection; DTT difficult to treat; * 3 polymicrobial cases each with a combination of a coagulase-negative *staphylococcus* and an *enterococcus.*

**Table 4 antibiotics-14-00752-t004:** Specific risk factors associated with each of the studied groups.

	Risk Factor	*p*-Value *	Odds Ratio	Confidence Interval
PJI DTT total	CRP at admission (≥92.1 mg/L)	0.001	6.981	1.367–35.63
	Hip joint	0.0225	3.478	0.361–33.538
PJI DTT Superinfection	Number of revisions (≥3)	<0.0001	1.288	1.100–1.508
	Chronic type of infections	0.0387	3.449	1.159–10.262

CRP, C-reactive protein; CCI, Charlson comorbidity index; * *p* < 0.05 indicates statistical significance.

## Data Availability

The data presented in this study is available on request from the corresponding author.

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
