# Peer review of "Prevalence and Risk Factors for Superinfection with a Difficult-to-Treat Pathogen in Periprosthetic Joint Infections"

_antibiotics, 2025, doi:10.3390/antibiotics14080752_

Round 1

Reviewer 1 Report

Comments and Suggestions for Authors

This is a well-structured and clinically relevant study that addresses an emerging challenge in orthopaedics: the role of difficult-to-treat (DTT) pathogens in periprosthetic joint infections (PJIs), particularly those arising as superinfections during treatment. The single-centre design with standardised surgical protocols is a strength, as it limits inter-institutional variability and improves internal validity.

However, several clarifications and additions are needed to enhance the manuscript:

  1. Please clearly define what is meant by "superinfection"—does this refer to the emergence of a completely new pathogen during the course of infection, or to resistance development in an existing pathogen?
  2. The study identifies clear high-risk factors for DTT PJI and superinfection. A brief discussion on practical preventive strategies—such as early initiation of broad-spectrum antibiotics, tailored surgical planning, or closer post-operative monitoring—would help improve the clinical applicability of the findings.
  3. Please expand on the clinical implications. How should these findings influence surgical or infectious disease practice? For example, should patients with high CRP levels and hip PJIs be considered for broader empirical antibiotic coverage or earlier transition to two-stage exchange? Clarifying this would significantly strengthen the manuscript's relevance to clinical decision-making.
  4. Infection-free survival is an important outcome in PJI research. Please include Kaplan–Meier curves or cumulative incidence plots stratified by DTT status, as this is standard in similar studies and would greatly enhance the impact and interpretability of your results.
  5. Ensure consistent use of English spelling conventions (e.g., "centre" vs "center") throughout the manuscript, in accordance with the journal’s style guide.
  6. Please define all abbreviations at first use, including ASA (American Society of Anesthesiologists score), to improve clarity for readers unfamiliar with these terms.

Comments on the Quality of English Language

Ensure consistent use of English spelling conventions (e.g., "centre" vs "center") throughout the manuscript, in accordance with the journal’s style guide.

Author Response

Dear Reviewer,

Thank you very much for allowing our publication to be revised.

Thank you for your constructive comments, which improved the quality of the manuscript substantially.

We have revised and corrected the text along the lines of your valuable suggestions.

In sum, we are of the opinion that both the quality and the explanatory power of our article have been significantly improved as a result of your comments. All the answers to the comments will be found below.

If you have any questions, we shall be at your disposal.

Kind regards,

The Authors

Reviewer 2 Report

Comments and Suggestions for Authors

Dear Authors,

I have reviewed your article entitled “Prevalence and Risk Factors for Superinfection with a Difficult-to-treat Pathogen in Periprosthetic Joint Infections”

Thank you for your efforts. Overall, it is well designed and written but, you should make a few revisions before it is accepted:

  • DDT should be DTT in line 32.
  • Enterobacterales’ should be written instead of ‘Enterobacteriaceae’ in line 50. Bacterial names should be written in italics.
  • DTT pathogens should be explained in more detail in the introduction section.
  • For the statistically significant situations in Table 2, the groups between which significance is found should be explained under the table. In this way, the table will be more understandable when reading.
  • The numbers in Table 3 are inconsistent. For example, PJI DTT Total is 78 people, but when the microorganisms that grow are added up, the number becomes 84. This should be corrected. In addition, is there a statistical difference between PJIs according to the microorganisms that grow? It should be stated.
  • Lines 127-131 ‘The remaining risk factors studied, such as age, sex, ASA, surgical treatment, antibiotic treatment, BMI, anticoagulation, failed to show statistically significant differences between the PJI DTT initial group and the PJI DTT superinfection group. A summary of the risk factors is shown in table 4.’ This statement should not be written as a negative statement. This finding is a valuable finding that shows that the groups are homogeneous in terms of sociodemographic characteristics.
  • It is recommended to add an exclusion criterion. For example, patients who are followed up for PJI but have no growth in culture is an exclusion an criterion.
  • Shouldn’t ‘Mean’ be used instead of ‘Median’ for ASA score in Table 1?
  • Most of the references are older than 5 years, and using references before 2017 should be avoided unless it is for basic information.

Best regards,

Author Response

(The authors gave the same response as above.)

Round 2

Reviewer 1 Report

Comments and Suggestions for Authors

The revised manuscript has been significantly improved. The authors have addressed the previous concerns appropriately, and the presentation is now much clearer. I believe the current version meets the journal’s standards and recommend it for publication.